# Mechanisms and Applications of Gastrointestinal Microbiota–Metabolite Interactions in Ruminants: A Review

**DOI:** 10.3390/microorganisms13122880

**Published:** 2025-12-18

**Authors:** Xingdong Wang, Huimin Wei, Gerelt Zhao

**Affiliations:** 1College of Life Science and Technology, Inner Mongolia Normal University, Hohhot 010022, China; 17621511689@163.com (U.); wxd17339929758@163.com (X.W.); yisahm@163.com (H.W.); 2Key Laboratory of Biodiversity Conservation and Sustainable Utilization in Mongolian Plateau for College and University of Inner Mongolia Autonomous Region, Hohhot 010022, China

**Keywords:** ruminants, gastrointestinal microbiome, metabolites, microbiota–host interaction, methane emissions

## Abstract

The gastrointestinal microbiota of ruminants constitutes a complex invisible organ, which converts plant fibers into volatile fatty acids (VFAs) and microbial protein through fermentation, serving as the primary energy and protein sources for the host. While substantial progress has been made in characterizing this system, critical gaps remain in understanding causal mechanisms and translating knowledge into scalable interventions. This review systematically synthesizes current knowledge on the composition, function, and metabolite profiles of gastrointestinal microbial communities in ruminants, with emphasis on interaction mechanisms, methodological advances, and intervention strategies. We highlight persistent challenges, including the uncultured majority of microbes, causal inference limitations, and translational bottlenecks. The review further evaluates strategies for targeted microbiome modulation aimed at improving production performance and reducing environmental emissions. Finally, we propose integrated research priorities for developing efficient, low-carbon, and sustainable ruminant production systems.

## 1. Introduction

Global livestock production faces the dual challenges of enhancing production efficiency and reducing environmental impacts. In this context, studying the ruminant gut microbiota provides new insights into addressing this dilemma [1]. Ruminants rely on their unique digestive system and symbiotic intestinal microbial communities to efficiently convert plant fibers, which are inedible to humans, and agricultural by-products into high-quality animal proteins (such as meat and milk). This process is crucial for ensuring global food security and alleviating resource competition between humans and livestock [2,3]. However, translating microbial ecology knowledge into practical, reliable, and economically viable strategies remains a major bottleneck. This review aims not only to describe the current state of knowledge but also to critically analyze the mechanistic evidence, identify persistent knowledge gaps, and evaluate the practical potential and constraints of proposed intervention strategies.

Metagenomic studies have revealed that the ruminant gut microbiome represents one of nature’s most efficient biotransformation systems. Among the more than 5800 microbial genomes identified, over 90% of the species remain uncultured, and their functions are largely unknown [4], This microbial dark matter constitutes a major frontier and challenge for functional research. The research value of this field is mainly reflected in three dimensions: production-maximizing microbial efficiency in converting low-value feed into high-value protein [5,6]; environment methane from ruminant microbial fermentation contributes approximately 5% of global anthropogenic greenhouse gas emissions, with a global warming potential 28 times that of carbon dioxide, rendering emission reduction critical for climate change mitigation [7,8,9,10]; and health discoveries regarding rumen microbial involvement in bile acid metabolism and early-life gut colonization provide new models for studying human metabolic diseases and infant gut development [11,12]. Despite this potential, studies often report contradictory findings regarding microbial responses to dietary shifts, and the extrapolation of in vitro or small-scale trial results to heterogeneous field conditions is problematic.

The ruminant gastrointestinal microbial community functions not merely in digestion but as an essential invisible organ [5]. Its metabolites profoundly impact the host: short-chain fatty acids (SCFAs, such as acetic acid, propionic acid, and butyric acid) are not only core energy substrates but also affect gene expression by regulating histone deacetylase activity; microbial derivatives of amino acids, such as tryptophan, can act as ligands to activate immune pathways and enhance intestinal barrier function [13]. Furthermore, microbially-mediated synthesis of B vitamins and vitamin K is essential for host metabolism and health [14]. A critical limitation is that most studies establish correlations rather than causality, and the precise concentration-dependent effects and tissue-specific actions of many metabolites in vivo are poorly quantified.

Innovations in research methodologies are central to progress in this field. From traditional culture methods to the combination of high-throughput technologies such as metagenomics, metabolomics, and stable isotope labeling, we can systematically analyze the structure and function of microbial communities and accurately track the transformation pathways of metabolites [15,16]. Nevertheless, each technology has inherent biases (e.g., PCR amplification bias, limitations in metabolite detection ranges, challenges in metagenome-assembled genome completeness) that must be considered when interpreting data. These technological advances have not only deepened our understanding of ruminant digestive physiology but also laid a foundation for developing microbiome-based precision feeding systems and transitioning animal production toward greater efficiency and environmental sustainability. Nevertheless, the field continues to face challenges in moving from descriptive phenomenology to mechanistic analysis and from single interventions to systemic regulation. This review systematically summarizes current research on microorganisms and metabolites in the ruminant gastrointestinal tract, focusing on the biosynthesis mechanisms and functional characteristics of microbial metabolites, elaborating their regulatory networks across host organs, critically evaluating existing intervention strategies, and outlining future research directions to advance novel microbiome-based strategies for ruminant health and production management.

## 2. Characteristics and Microbial Ecology of the Ruminant Digestive System

### 2.1. Species Diversity and Digestive System Characteristics

Ruminants are an important group in the mammalian artiodactyla, with a wide range of species, including Cattle, Sheep, Deer and Camelidae [17]. Globally, there are over 3.5 billion cattle, sheep, and goats, which form the backbone of animal husbandry [18]. Different ruminants have significant differences in body size and ecological adaptability. Species range from the 2 kg lesser mouse deer (*Tragulus kanchil*) to the 1.5 ton African buffalo (*Syncerus caffer*). This includes heat-tolerant buffaloes in tropical wetlands and cold-adapted yaks (*Bos mutus*) on the Qinghai–Tibet Plateau, all demonstrating remarkable ecological adaptability [19,20]. This diversity remains underexplored in microbiome research, which has disproportionately focused on a few commercial breeds, limiting our understanding of adaptive microbiome traits.

Most ruminants have a complex stomach system composed of the rumen, reticulum, omasum and abomasum, in which the rumen is the core site of microbial fermentation [21]. As pseudo-ruminants, camels have a three-chambered stomach structure (rumen, reticulum, omasum + abomasum) distinct from other ruminants, reflecting a unique evolutionary adaptation [22,23]. The rumen microbial communities of key economic animals such as cattle, sheep, goats and yaks have their own characteristics, which are closely related to their dietary habits and environmental adaptability, and provide an ideal model for studying host–microorganism co-evolution [24,25,26]. However, comparative studies across these species to disentangle the effects of phylogeny, diet, and environment on microbiome assembly are still limited. Table 1 summarizes the distribution and functional characteristics of microbial communities in the digestive tract of ruminants, providing a microbiological basis for understanding ruminant digestive physiology.

### 2.2. Metabolic Characteristics of Ruminants

Compared with non-ruminants (monogastric animals), the entire metabolic activity of ruminants is almost entirely centered on the huge microbial ecosystem in their rumens, forming a series of unique metabolic characteristics.

#### 2.2.1. Fundamental Changes in Energy Metabolism

Rumen microorganisms produce volatile fatty acids (VFAs), mainly including acetic acid, propionic acid and butyric acid, by fermenting carbohydrates (mainly cellulose and hemicellulose) in the diet. While cellulose and hemicellulose are key fermentable substrates, ruminants exhibit superior fiber fermentation capacity compared to monogastric species like pigs and poultry. It should be noted, however, that some monogastric species such as equines and rabbits can also ferment fiber appreciably in their ceca. After being absorbed by the rumen wall, these VFAs can meet 70–80% of the maintenance and growth energy needs of ruminants. Thus, VFAs replace glucose as the primary energy source for ruminants, a fundamental difference from monogastric animals directly digesting and absorbing glucose as the core energy [5,34].

#### 2.2.2. Complex Hydrogen Metabolic Network and Methane Production

There is an extremely complex hydrogen (H_2_) metabolic network in the anaerobic environment of the rumen. Metagenomic data show that about half of the rumen microbial genome encodes hydrogenase, which can release hydrogen during the decomposition of cellulose. Meanwhile, *methanogens*—a specialized group of archaea—utilize hydrogen and carbon dioxide (CO_2_) as substrates to produce methane (CH_4_). Although this interspecies hydrogen transfer process maintained a low hydrogen partial pressure in the rumen and ensured the continuous activity of cellulolytic bacteria, it also led to the loss of 2–12% of gross energy in the feed in the form of methane. More importantly, methane, as a potent greenhouse gas, constitutes one of the main sources of greenhouse gas emissions from animal husbandry and has caused widespread environmental concerns [35,36,37,38].

#### 2.2.3. Efficient Nitrogen Utilization and Transformation

The nitrogen metabolism of ruminants shows excellent flexibility and efficiency. Rumen microorganisms can utilize non-protein nitrogen (NPN), such as urea added to feed or endogenous urea circulating in the body, to convert it into high-quality microbial crude protein (MCP). These MCPs are then digested and absorbed in the abomasum and small intestine, providing 40–80% of the total protein absorbed by the small intestine for ruminants. This mechanism allows ruminants to efficiently utilize low-quality nitrogen sources or non-protein nitrogen sources to reduce nitrogen emissions while reducing feeding costs, while monogastric animals are highly dependent on high-quality pre-formed protein in the diet [39,40].

#### 2.2.4. Unique Biotransformation and Detoxification Capabilities

The rumen microbial community gives ruminants the extraordinary ability to treat a variety of plant secondary metabolites. For example, they can degrade the anti-nutritional factor-gossypol in cottonseed, thereby reducing its toxicity and making cottonseed meal a useful feed raw material. In addition, rumen microorganisms can not only partially degrade tannins that are widely present in plants, but also promote the formation of stable complexes between tannins and feed proteins. This complex is not easily degraded in the rumen, thereby protecting the protein, enabling it to flow more to the abomasum and small intestine to be effectively absorbed, and improving the utilization efficiency of essential amino acids. These special metabolic capacities are extremely limited or completely absent in monogastric animals [41,42,43].

#### 2.2.5. Development Stage Characteristics

The rumen of newborn ruminants (such as calves, lambs, and kids) is not mature in anatomical structure and function. At this time, its digestive physiology is closer to that of monogastric animals, mainly relying on lactose and fat in breast milk to provide energy. As the animals began to eat solid feed, the rumen volume increased rapidly, the papillae on the rumen wall developed, and the microbial community began to undergo dramatic succession. Usually after 6–8 weeks of age, a functional stable adult model microbial community with Firmicutes and Bacteroidetes as the absolute dominant flora is gradually established, marking the official transformation of animals into typical ruminants [44,45]. This metabolic transition period from a monogastric-like state to a mature ruminant state is a key developmental stage unique to ruminants, and the nutrition and management during this period profoundly impact their lifelong production performance.

## 3. Mechanism of Interaction Between Microbiome and Metabolome

The microbial community in the intestinal tract of ruminants and the host metabolic system form a precise synergistic network. Bioactive molecules such as short-chain fatty acids, bile acid derivatives, and vitamins produced by microbial metabolism play key roles in energy supply and nutrient metabolism. These metabolites not only act as energy substrates but also affect the host’s physiological state by regulating immune system function and metabolic pathway activity. It is noteworthy that changes in the host’s metabolic environment also exert selective pressure on the composition of the microbial community, promoting adaptive changes in microbial population structure. This bidirectional regulation mechanism establishes a symbiotic relationship of dynamic balance between microorganisms and the host, which decisively influences the nutritional metabolic efficiency, energy conversion processes, and overall health status of ruminants. While this framework is widely accepted, the relative contribution of host genetics versus microbial activity to metabolic phenotypes, and the precise molecular triggers for microbial community shifts, are areas of active debate and require more causal evidence. This provides a new research paradigm for further analyzing the co-evolutionary relationship between hosts and microorganisms.

### 3.1. Microorganism-Mediated Metabolic Regulation

The microbiome degrades cellulose, amino acids, and lipids to produce key metabolites (e.g., SCFAs, bile acids, vitamins), supplying up to 70% of the host’s energy [46,47]. *Prevotella bryantii*
*B14* activates the GPR109A–mTORC1 signaling pathway in the mammary gland via nicotinic acid metabolism, thereby regulating milk fat synthesis and revealing the mechanism underlying the rumen–blood–mammary axis [48]. In addition, virus-encoded auxiliary metabolic genes (AMGs) can expand host metabolic function and optimize nutrient utilization efficiency [49].

### 3.2. The Feedback Effect of Metabolites on Microorganisms

Microbial metabolites have a significant feedback regulation effect on microbial community structure. The accumulation of metabolites such as short-chain fatty acids and lactic acid can drive changes in the structure of the flora. For example, high-starch diets promote the enrichment of Succinivibrionaceae and change the flow of hydrogen metabolism, thereby affecting methane emissions [50,51]. The concentration of metabolites showed a dual effect. The appropriate concentration of butyrate promoted the development of rumen epithelium, while excessive butyrate led to epithelial parakeratosis [52].

### 3.3. Host–Microorganism Metabolic Network

Host genetic background and microbiome together constitute a synergistic metabolic network. Bayesian model analysis showed that the host genetic factors explained about 24% of the variation in methane emissions, and the microbiome contributed about 7% [7]. Different intestinal types (such as TR and CO) have different microbial interaction patterns. TR intestinal type shows a more complex interconnection network and promotes efficient substrate utilization [48]. The interpretation of such network analyses requires caution, as correlation does not imply causation, and network stability and functional redundancy are rarely assessed.

### 3.4. Temporal and Spatial Distribution of Microbial Metabolites

Microbial metabolic activities showed significant spatial and temporal dynamic characteristics. In the rumen, 61.99% of the bacteria and 66.93% of the protozoa showed a circadian rhythm synchronized with the feeding time. The distribution of mobile genetic elements (MGEs) in the stomach and small intestine is characterized by high at both ends and low in the middle. Among them, the stomach is rich in CAZyme-carrying elements, and the ileum has the most resistance genes [53].

### 3.5. Host Immune Metabolic Regulation

Microbial metabolites affect host health through immune regulation. Short-chain fatty acids promote the formation of anti-inflammatory immune cell phenotype and enhance intestinal barrier function by inhibiting HDAC activity and activating GPCR pathway [54]. Microbial tryptophan metabolite 3-indoleacetic acid (IAA) alleviates subacute rumen acidosis-related inflammation by inhibiting Th17 cell differentiation and IL-17 signaling pathway [55].

## 4. Ruminant Gastrointestinal Metabolite System

The intestinal microbial community of ruminants produces a variety of bioactive substances through complex metabolic pathways. These metabolites can be broadly categorized into primary and secondary metabolites based on their synthesis pathways and fundamental functions. They play key roles in host physiological processes. However, quantifying their absolute concentrations in different gut compartments and understanding inter-individual variation remains challenging. Table 2 systematically summarizes the types, producing flora, functional characteristics, and regulatory mechanisms of major intestinal metabolites in ruminants.

### 4.1. Primary Metabolites and Their Functions

Primary metabolites are small molecules synthesized directly by microorganisms and host cells as part of fundamental life processes. They are mainly involved in energy metabolism and cell construction, in contrast to secondary metabolites (usually with signal or defense functions). They are the building blocks and energy currency of cell construction, directly participating in and driving the basic physiological functions of the host [64]. These substances mainly include amino acids, volatile fatty acids (VFAs), organic acids (such as lactic acid and succinic acid), and nucleotides. Their concentration and dynamic balance in organisms are the core biochemical indicators for evaluating the nutritional status, health level and production performance of ruminants [65].

Among the many primary metabolites, volatile fatty acids (VFAs), which are produced by carbohydrate fermentation by rumen microorganisms, are undoubtedly the largest and most important metabolite group in the intestine [66]. According to its carbon chain skeleton structure, VFAs can be mainly divided into straight-chain VFAs (straight-chain fatty acids) and branched-chain VFAs (branched-chain fatty acids, BCFAs). Linear VFA mainly includes acetic acid (acetate), propionic acid (propionate), and butyric acid (butyrate). They are the main end products of carbohydrate fermentation and the most important energy source for ruminants. The carboxyl (-COOH) functional group at the molecular end gives them weak acidity and good water solubility, which enables them to be efficiently absorbed into the blood circulation through rumen epithelial cells [67]. Branched-chain VFAs (BCFAs), such as isobutyrate and isovalerate, are mainly produced by protein-degrading bacteria (such as Clostridium) that decompose branched-chain amino acids (valine, leucine, isoleucine) in feed. The methyl branch on the carbon chain forms a unique spatial conformation, which is essential for maintaining the growth and activity of certain fiber-degrading microbial populations (such as certain rumen cocci) and plays a role similar to growth factors [68].

In addition to VFAs, amino acids and their derivatives produced by rumen microorganisms that degrade feed proteins are also important primary metabolites. The specific functional group structure of these amino acids, such as the sulfur atom in the sulfur-containing amino acid (methionine, cysteine) molecule, or the benzene ring structure in the aromatic amino acid (phenylalanine, tyrosine) molecule, directly determines their subsequent pathways and final products for further metabolism by microorganisms, such as conversion to VFA or synthesis of new microbial proteins [69,70].

In addition, some intermediate products in the process of microbial sugar metabolism, such as dicarboxylic acids such as succinic acid (Succinate) and fumarate (Fumarate), and biogenic amines (such as histamine and tyramine) produced by amino acid decarboxylation, constitute an important acid–base buffer system in the rumen. They can neutralize a large amount of acidic substances produced during the fermentation process, which plays an indispensable key role in maintaining a relatively stable pH environment inside the huge fermentation tank and ensuring the coordinated growth of various microorganisms and the normal conduct of fermentation activities [71,72].

### 4.2. Secondary Metabolites and Their Activities

The secondary metabolites produced by ruminant intestinal microorganisms are the key mediators to cope with environmental stress and maintain host interaction. Different from the primary metabolites directly involved in growth, secondary metabolites are mainly responsible for immune regulation, signal transduction and ecological balance. Their synthetic pathways are complex, and their biological activities are significant.

The secondary metabolites have a wide range of sources, and the functions are category-specific: (1) microbial synthesis: such as B vitamins (pyridoxal, cobalamin) and antimicrobial peptides (such as Gassericin A produced by rumen lactic acid bacteria), the latter has a strong inhibitory effect on pathogenic bacteria [73]; (2) microbial transformation: represented by secondary bile acids, primary bile acids are modified by intestinal flora to participate in lipid digestion and signal transduction [74]; (3) plant-derived: tannins, flavonoids and other plant secondary metabolites in the diet still retain biological activity after microbial modification.

Secondary metabolites affect the host through multiple pathways: (1) Immune and barrier maintenance: Gassericin A enhances anti-diarrhea ability by acting on epithelial cell membrane protein KRT19 and reducing intracellular cAMP/cGMP levels [5]; tryptophan metabolites (such as indole derivatives) promote the secretion of IL-22 and strengthen the immune barrier by activating aromatic hydrocarbon receptors [75]. (2) Energy metabolism and methane mitigation: Tannins and flavonoids (such as quercetin) can specifically inhibit the activity of methanogenic archaea. Quercetin can reduce methane production by up to 43% in vitro [76]. However, in vivo efficacy is often lower due to ruminal degradation, adaptation of microbial communities, and dose limitations. (3) Nutrient metabolism: Vitamin B family, as a key coenzyme, is involved in a variety of metabolic pathways, among which cobalt amine (B_12_) is completely dependent on rumen microbial synthesis [14].

The synthesis of secondary metabolites is regulated by dietary composition, host genetics and microbial community. The addition of plant polyphenols or functional oils to the diet can reshape the flora structure and change the metabolic profile [77]. Due to their unique evolutionary history and feeding habits, the rumen metabolite profiles of different ruminant species show significant species specificity. For example, camels are rich in organic acids, goats are mainly composed of alcohols and hydrocarbons, sheep accumulate indoles, and cattle are rich in sesquiterpenoids, which profoundly reflects the unique metabolic characteristics formed by the long-term co-evolution of hosts and microorganisms [78].

## 5. Research Methods and Technical Progress

### 5.1. Microbial Culture Technology

Traditional culture techniques, especially the Hungate strict anaerobic system, form the basis for the isolation and preservation of rumen functional microorganisms (such as *Fibrobacter succinogenes*, *Ruminococcus flavefaciens*), but only about 3.6% of rumen microorganisms can be cultured, with significant limitations [16,79]. In order to break through this bottleneck, new training technology came into being. Culturing omics (such as iChip in situ culture) and single-cell technology (such as single-cell sequencing) have successfully obtained the genomes of a variety of difficult-to-culture microorganisms [80,81]. The microfluidic system can accurately simulate the rumen microenvironment and realize the tracking of physiological activities at the single cell level, which greatly expands our understanding of microbial life processes [82].

### 5.2. Integration of Molecular Biology and Omics

Early microbial community analyses historically relied on PCR and DGGE targeting 16S rRNA genes, but these methods have limited resolution and are increasingly supplanted by high-throughput sequencing [83,84]. The application of high-throughput sequencing technology (such as Illumina platform) further revealed the fine differences in rumen microbial community structure and function under different breeds (such as Bohai black cattle and Holstein cattle) and dietary conditions.

A single omics technology is not enough to reveal complex interaction mechanisms, and multi-omics integration (genome, transcriptome, metabolome, etc.) has become an inevitable trend. By integrating metagenomic, metabolic, and liver transcriptome data, studies have confirmed that hindgut microbial imbalance can cause energy metabolism disorders by inhibiting the liver AMPK-PPARα axis [85]. Standardized analysis processes (such as QIIME2 (2024.10), MetaCyc (28.1)) and bioinformatics tools (such as antiSMASH (8.0), BiG-SCAPE (v1.1.2)) provide strong support for the analysis of massive data and the reconstruction of microbial metabolic networks [86,87]. Nevertheless, these approaches possess inherent limitations. PCR amplification bias, incomplete metagenome-assembled genomes (MAGs), and variability in DNA extraction protocols can affect reproducibility. Furthermore, predictive metabolic models often lack experimental validation, and integrating multi-omics data remains computationally challenging.

### 5.3. Metabolomic Analysis Techniques

Metabolomics is the core tool for analyzing host-microorganism interactions. Targeted metabolomics focuses on specific metabolic pathways, and with high sensitivity and specificity, accurate quantification of key metabolites (such as gossypol isomers, volatile fatty acids) can be achieved [88]. Non-targeted metabolomics analyzes all metabolites in a sample without bias, and can systematically reveal the panorama of the metabolic network, such as the successful identification of differential metabolites in the feces of the ketosis group and the healthy group [85], or the significant effect of the intensive feeding system on the plasma metabolic profile of yaks [89].

Chromatography–mass spectrometry (LC-MS/GC-MS) is the backbone of the above strategy. Its high resolution and sensitivity provide a reliable guarantee for complex biological sample analysis. In order to further improve the analysis dimension, in situ detection techniques (such as MALDI-MS spatial imaging and SERS technology) have realized the visualization of the spatial distribution of metabolites in tissue sections and rapid screening of body fluids, providing a new perspective for studying the dynamics of intestinal microbial metabolites [90,91].

Dynamic tracing technology has promoted the transition of metabolic research from “static description” to “dynamic analysis”. Stable isotope labeling technology (such as ^13^C and ^15^N) is the core of it, which can accurately track the transformation trajectory and cross-tissue flux of metabolites in vivo, and provide key data for quantifying rumen microbial-host co-metabolism [92,93].

Four-dimensional metabolomics (LC-IM-MS) is a major breakthrough in metabolic analysis. Through the combination of liquid chromatography (LC), ion mobility (IM) and mass spectrometry (MS), it effectively solves the recognition problem of isomers and greatly improves the resolution ability of secondary metabolites with similar structures in ruminants [94]. When this technique is combined with stable isotope labeling, the transformation trajectory of metabolites can be fully revealed in the spatial and temporal dimensions. Combined with metabolic flux analysis (MFA) and multi-omics integration, key biological pathways can be systematically elucidated [93].

Metabolomics has become an indispensable tool to reveal the complex interaction mechanism between ruminant gut microbes and hosts, showing unique advantages. Table 3 summarizes the characteristics of major metabolomics analysis technology platforms, providing an important technical reference for selecting metabolite detection methods and techniques in ruminant nutrition research.

## 6. Influencing Factors

The interaction mechanism of ruminant gut microbiome and metabolome is a complex dynamic process, involving the synergistic regulation of multi-dimensional factors. This regulatory network not only determines the ecological function of the microbial community but also directly affects the metabolic health and production performance of the host. However, the relative importance of these factors and their interactions are often context-dependent and not fully understood, leading to variable outcomes in different production settings.

### 6.1. Dietary Factors

Dietary composition is a key factor in regulating the structure and function of the microbiome. Altering the concentrate-to-forage ratio can significantly shift rumen fermentation patterns; for example, increasing the proportion of concentrate reduces the acetate-to-propionate ratio [97]. High-fiber feed promotes the proliferation of cellulolytic bacteria, and the methane energy conversion rate can reach more than 10% of the total energy of the diet. However, high-grain feed (>80%) reduced the methane energy conversion rate to 3–4%. Feed additives such as ionophore antibiotics, tannins, and tea saponin can optimize fermentation efficiency and reduce methane emissions by regulating microbial enzyme activity and inhibiting methanogens [96,98,99]. The long-term efficacy and potential for microbial adaptation or resistance to some additives, like ionophores, require careful monitoring. Maternal dietary intervention can affect the microbial community structure and function of offspring through cross-representative genetic or microbial vertical transmission.

### 6.2. Environmental Factors

Environmental conditions affect microbial metabolic activity by changing the rumen microenvironment. A decrease in temperature can promote the transformation of fermentation mode to propionic acid and reduce methane synthesis [100]. An increase in humidity will lead to a decrease in rumen pH and significantly inhibit the activity of methanogenic archaea [97]. Altitude changes affect the redox potential, and the cellulose decomposition rate of strictly anaerobic bacteria increases by about 45% when Eh decreases [101]. In addition, stress factors such as transportation can disrupt the balance of microbial communities and affect host health and production performance [102]. These factors often interact; for instance, dietary changes during heat stress can compound microbial dysbiosis.

### 6.3. Host Factor

The host genetic background has significant selection pressure on the microbiome. The heritability of methane emission in Holstein cows is 0.12–0.45, indicating that this trait has a clear genetic basis [7]. The mGWAS study showed that the LCT locus was associated with bifidobacteria abundance [103]. Different physiological stages such as age, pregnancy and lactation cycles can also significantly change the microbial community structure. For example, milk fat synthesis in high-yield dairy goats is closely related to the activation of nicotinic acid metabolic pathway [48]. The differential expression of host immune genes and the change of metabolite content jointly regulate the microbiota–host interaction network [104]. Substantial inter-individual variation exists even within genetically similar herds under uniform management, highlighting the role of stochastic assembly and early-life events in shaping the mature microbiome. Table 4 summarizes the effects of different influencing factors on the key indicators of rumen fermentation. These changes further change the type and concentration of microbial metabolites, thereby affecting the host’s energy metabolism and immune regulation function.

## 7. Control Strategy

The regulation of the interaction between the ruminant microbiome and metabolome has led to the development of a multi-strategy synergistic intervention system, covering nutritional regulation, microbial intervention, biotechnology application, behavioral phenotype screening, and epigenetic regulation. This provides important pathways for improving production performance and achieving environmentally friendly animal husbandry. Future regulation strategies should be based on systematic thinking, simultaneously evaluating their comprehensive effects on production performance (e.g., milk fat percentage, daily weight gain) and environmental indicators (e.g., methane, nitrogen, and phosphorus emissions) to achieve synergy between yield increase and emission reduction. A critical barrier is the frequent trade-off between controlled-experiment efficacy and the stability, cost-effectiveness, and practicality required in diverse commercial systems.

### 7.1. Nutritional Regulation Strategy

By optimizing the diet structure and supplementing key nutrients, the microbial metabolic network can be effectively reshaped. Adjusting the NFC/NDF ratio can improve rumen fermentation and reduce methane emissions [107]. Medium-chain fatty acids and polyunsaturated fatty acids (dietary lipid < 6%) can reduce methane emissions by up to 20% by inhibiting methanogens and competing hydrogen sources, respectively [108]. The addition of fumaric acid can reduce the methane synthesis substrate by consuming hydrogen ions [109]. Plant secondary metabolites (such as tannins and citrus flavonoids) have anti-inflammatory effects while inhibiting methane production [110]. Maternal dietary intervention is a potential strategy to shape the microbiome of offspring and improve their lifelong production performance [111]. The economic viability of sustained additive supplementation and potential negative impacts (e.g., on fiber digestion or milk fat) require thorough evaluation.

### 7.2. Microbial Intervention Strategies

Probiotics (such as *Lactobacillus*, *Saccharomyces cerevisiae*) and specific functional strains (such as *Prevotella bryantii B_14_*) can improve rumen function, promote milk fat synthesis and inhibit methane production [48]. Methane inhibitors (such as 3-NOP) and phage therapy can specifically target *methanogens* [25,112]. Compound microbial agents (such as *Mortierella*, *Schizochytrium*) can increase milk fat rate and polyunsaturated fatty acid content [113]. Fecal microbiota transplantation technology can repair the intestinal barrier and improve the health of calves with passive immune failure [114]. Key challenges include ensuring survival/engraftment in the competitive rumen, regulatory hurdles for microbials/phages, and consumer acceptance of FMT.

### 7.3. Biotechnology

Gene editing techniques (such as CRISPR/Cas) can control methane emissions at the source by modifying forage quality-related genes or key metabolic genes of *methanogens* (Mcr, mtmCB, etc.) [115]. Metabolic engineering and synthetic biology strategies have shown great potential. For example, heterologous expression of highly efficient cellulase genes or design and synthesis of microbial communities are expected to directly improve the degradation efficiency of roughage in the rumen [116]. Metabolic engineering technology can be used to regulate the expression of key genes of methanogens or optimize the metabolic efficiency of rumen bacteria (such as *fiber-degrading bacteria*) [117]. The establishment of yak fecal bacterial genome reference set (YFR) provides a resource basis for target screening [118]. The rumMGE database system identified five types of mobile genetic elements in the digestive tract of ruminants, of which 96.5% phages were newly discovered species, providing new tools for precise regulation [53]. These approaches face significant technical, ethical, and regulatory hurdles. The ecological consequences of releasing engineered organisms demand rigorous, long-term assessment.

### 7.4. Screening Based on Behavioral Phenotype

Behavioral traits such as rumination time (RT) are closely associated with microbiome function and methane emissions. The concentration of total volatile fatty acids in the rumen of cows with long rumination time was higher, and the number of gene copies involved in the alternative H_2_ utilization pathway increased, prompting hydrogen to flow to the non-methanogenic pathway, and methane emissions decreased by 26% [119]. Wearable device technology enables the combination of rumination time monitoring and microbiome analysis to provide a feasible solution for the screening of low methane emission phenotypes [120]. This strategy is promising for genetic programs but requires validation across diverse breeds, diets, and environments.

### 7.5. Epigenetic-Based Regulation

Microbial metabolites (such as short-chain fatty acids) can regulate host gene expression by inhibiting epigenetic mechanisms such as histone deacetylase (HDAC) [121]. Butyrate can induce histone acylation and promote intestinal development and immune balance [122]. Microbial synthesis of folic acid can promote the process of DNA methylation in the digestive tract. There are conserved methylation regions among ruminant species, which may be related to host-microbe interaction [123]. The heritable characteristics of epigenetic modifications provide the possibility for cross-generational regulation. This area is largely descriptive in ruminants; causal evidence linking specific metabolites to defined epigenetic changes and durable phenotypic outcomes is needed.

### 7.6. Limitations and Practical Considerations of Intervention Strategies

While numerous strategies show promise in controlled settings, their translation to heterogeneous field conditions faces significant hurdles. Microbial adaptability, host genetic variability, and environmental interactions often attenuate intervention efficacy. Economic feasibility, regulatory approval, and consumer acceptance further constrain implementation. For instance, long-term use of feed additives may lead to microbial resistance or adverse effects on fiber digestion. Similarly, genetic editing and synthetic biology approaches raise ethical and biosafety concerns. Future research must prioritize robustness, scalability, and systemic impact assessment.

## 8. Challenges and Prospects

Research on the ruminant gut microbiome is in a critical transition period from describing phenomena to analyzing mechanisms, from single intervention to systematic regulation. At present, there are three core challenges; the dilemma of functional analysis of difficult-to-cultivate microorganisms at the technical level; the application level faces the stability and security challenges of the regulation strategy; the cognitive level is limited by the research limitations of economic species and the lack of overall ecological perspective. In order to break through these limitations, it is urgent to construct a new systematic research paradigm that integrates full pedigree-cross-scale-intelligent approaches.

### 8.1. Expanding the Breadth of Research: Building a Full-Spectrum Microbial Resource and Integrated Ecosystem

While most current research focuses on economically important species such as cattle and sheep, the microbiomes of many wild and rare ruminants remain poorly characterized [124]. In the future, a metagenomic project covering the entire ruminant evolutionary tree should be initiated to systematically analyze the microbial resources of extreme environmental adaptation species, and develop a new direction of conservation microbiome to provide new strategies for the health maintenance of endangered species [35,125]. At the same time, it is necessary to establish a cognitive framework of integrated microbiome from pasture to gut, and regard environmental microbiome (such as soil, feed, water source) as the seed banks and regulatory source of gut microbiota, so as to further explore its cross-niche transmission and interaction rules [125,126]. In addition, the functional study of hindgut microbiome should be emphasized to clarify its role in overall energy metabolism and health maintenance.

### 8.2. Deepening Research Depth: From Correlation Analysis to Causal Mechanism and Ecological Theory

The current research needs to break through the three major knowledge gaps: the differences in the core metabolic networks of different ruminant groups, the coupling mechanism of microbial function and ecological factors, and the causal functional interaction behind the association phenomenon [127,128]. In order to fill these gaps, future research should focus on: strengthening cross-species comparative research and environmental microbiome, combining stable isotope tracing and targeted metabolomics to accurately quantify cross-species metabolic flux [129]; the theory of microbial ecology was introduced to reveal the universal laws of community assembly rules, succession dynamics and ecosystem resilience. Using sterile animal models, gene editing and other techniques, we demonstrated the causal path of microbial-host interaction at the molecular and cellular levels, and finally constructed a environment-microbiome-host adaptability prediction model [130].

### 8.3. Innovative Research Methods: Breaking Through Technical Bottlenecks and Driving Data-Model Fusion

Although multi-omics technology has been widely used, it still faces the challenges of difficult-to-cultivate microbial analysis, data integration standardization and deep application of artificial intelligence. Particularly important is the difficulty in directly comparing and reproducing many research results due to inconsistencies in animal individual differences, feeding management, sample processing, and analytical procedures. Establishing standardized operating procedures (SOPs) and high-quality public databases is crucial. The future innovation path should be a two-wheel drive: (1) Breakthrough in cognitive technology: focusing on in situ functional analysis. For example, Single-cell multi-omics integration to link genetic potential with metabolic activity in uncultured microbes. The spatial multi-omics and organoid co-culture model was used to visualize and verify the interaction mechanism of the microbial–mucosal interface at the tissue level [131,132,133]. (2) Data and model driven: promoting the construction of a global data alliance and standardized database ecosystem [134]; AI-driven predictive modeling of microbiome responses to dietary or environmental shifts and building a digital twin of ruminants system to provide the algorithmic core for precision nutrition [135,136]; by introducing the concept of synthetic biology, designing functional synthetic microbial communities, and simultaneously establishing a strict ecological security assessment system, we investigated its long-term impact on non-target communities, and achieved a leap from “interpreting nature” to “safely designing function” [130,137].

The transformation of this research paradigm will promote the research of ruminant microbiome from the traditional “production orientation” to the new stage of “ecology–production-health–design” multi-coordination. It not only provides core scientific and technological support for the sustainable development of animal husbandry but also offers a new perspective for understanding host–microorganism co-evolution, and ultimately leads the field into a new era of predictable, designable and controllable microbiome precise regulation [2].

## 9. Conclusions

This review synthesizes the intricate relationships between the gastrointestinal microbiota, their metabolites, and the ruminant host, highlighting both remarkable progress and persistent challenges. Key advances include genomic cataloging of rumen microbes, elucidation of major metabolic pathways (VFAs, hydrogen, and nitrogen), and the recognition of microbial metabolites as central regulators of host energy metabolism, immunity, and health. The development of multi-omics and stable isotope tracing has transformed our observational capacity.

However, critical knowledge gaps remain. First, the field is rich in correlations but poor in causal mechanisms. The functional roles of the vast uncultured majority and the causal links between specific microbes, metabolites, and host phenotypes require rigorous validation using gnotobiotic models and synthetic communities. Second, research is imbalanced, heavily focused on a few commercial species and the rumen, neglecting the hindgut’s role and the microbial adaptations of wild and indigenous ruminants. Third, translating promising interventions into robust, scalable, and economically viable on-farm strategies is a major bottleneck. Issues of microbial community stability, host adaptation, long-term efficacy, cost, and regulatory approval hinder implementation.

Future research must prioritize a holistic and mechanistic approach. We recommend (1) expanding comparative studies across the ruminant phylogeny and integrating environmental microbiome data to understand fundamental principles of microbiome assembly and adaptation; (2) employing causal inference tools—germ-free models, temporal multi-omics, metabolic flux analysis—to move beyond association and define mechanism; (3) developing and validating next-generation interventions, including precision probiotics, engineered enzymes/microbes (with thorough safety assessments), and epigenetic modulators from microbial metabolites; and (4) embracing digital agriculture by integrating real-time sensor data (rumination, methane) with microbiome analytics for phenotyping and precision management.

Ultimately, harnessing the ruminant microbiome for sustainable production requires a paradigm shift from descriptive ecology to predictive biology and safe engineering. By addressing these priorities, targeted strategies can be developed to simultaneously enhance productivity, improve health and welfare, and mitigate environmental impact, contributing to a more resilient and sustainable global food system.

## Figures and Tables

**Table 1 microorganisms-13-02880-t001:** Distribution and functional characteristics of microbial communities in the digestive tract of ruminants.

Digestive Tract Parts	Dominant Flora/Functional Flora	Cell Density/Abundance	Main Functions	Influencing Factors
Rumen	*Prevotella*, *Ruminococcus*, *Butyrivibrio* [27]	10^10^–10^11^ cells/mL [27]	Fiber degradation, nitrogen metabolism, and volatile fatty acid synthesis [27]	Diet composition, host genetics [27,28]
Rumen	*Fibrobacter succinogenes*, *Ruminococcus albus* [29]	-	Degradation of cellulose and hemicellulose [29]	Dietary fiber content [29]
Rumen	*Methanobrevibacter* [27]	It accounts for 0.3–3% of the microbiome [27]	methane production [27]	hydrogen availability [27]
Small intestine	*Lactobacillus*, *Streptococcus*, Enterobacteriaceae, etc. [30]	-	Oligopeptides and amino acid absorption, subsequent digestion of residual starch and protein, immune regulation [30]	Amino acid composition [30]
Large intestine	*Bacteroides*, *Clostridium*, *Fibrobacter,* etc. [31,32]	-	Undigested fiber and protein fermentation, SCFAs secondary synthesis, water absorption [32]	Dietary residues [32]
Rumen liquid phase	*Succinivibrio dextrinosolvens* [33]	-	Pectin and maltose are degraded to produce succinic acid and acetic acid [33]	soluble carbohydrates [33]
Rumen solid phase	*Fibrobacter succinogenes* [29]	-	Crystalline cellulose degradation [29]	fiber substrate [29]
ruminal epithelium	-	-	Epithelial cell regulation [27]	host-microorganism interaction [27]
Digestive tract of newborn individual	Mother-derived microorganisms [31]	-	Early colonization [31]	Maternal exposure and environmental exposure [31]

**Table 2 microorganisms-13-02880-t002:** Metabolites and functional characteristics of gut microbiota in ruminants.

Metabolite Type	Mainly Producing Flora	Main Functional Characteristics	Concentration/Ratio Characteristics	Key Regulatory Mechanisms
Short-chain fatty acids (SCFAs)	Firmicutes (butyric acid), Bacteroidetes (acetic acid/propionic acid) [56]	Energy supply (meeting 50% of rumen energy demand) [57], immune regulation (GPR41/43 pathway) [58], intestinal barrier function	Proximal large intestine 57:22:21 (acetic acid:propionic acid:butyric acid) [59]	Activation of G protein-coupled receptors, inhibition of HDAC [13], inhibition of NF-κB signaling [60], and promotion of Treg cell proliferation [61]
Bile acid	*Clostridium* (such as *Clostridium scindens*) [46,55]	lipid digestion and absorption, GLP-1 secretion promotion [59], metabolic regulation (FXR/GPBAR1 receptor) [12]	High-starch diet increases TCDCA/TDCA levels [62]	7-dehydroxylation transformation [56], FXR nuclear receptor activation [17], TGR5 receptor-mediated energy expenditure [59]
Tryptophan derivatives	*Bifidobacterium* et al. [56]	Immunomodulatory (IL-22 induction) [12], neurotransmitter synthesis (5-HT) [63], inflammatory regulation (Th17 inhibition) [55]	Increased metabolic activity in IBD patients	AhR receptor activation [22], IDO enzyme expression upregulation [56], KYN pathway metabolite regulation [12]
Polyamines	*Prevotella*, *Ruminococcus*	Epithelial cell proliferation promotion, autophagy regulation, and anti-inflammatory effects (IL-10 promotion/TNF-α inhibition)	It accounts for 30–40% of intestinal polyamine pool	Arginine decarboxylase/ornithine decarboxylase pathway, HDAC inhibition, macrophage polarization regulation
Vitamins	Bacteroidetes, Firmicutes	Nutrition supply (B vitamins), coagulation function (VitK), immune regulation (Foxp3+ T cell homeostasis) [56]	Nicotinic acid inhibits IL-8 production [12]	Epigenetic modification is involved in the regulation of inflammatory signaling pathways
Choline metabolites	Specific intestinal flora	Cardiovascular health indicators, lipid metabolism regulation	The content of TMA decreased in IBD patients [12]	Choline-TMA-TMAO metabolic axis and liver transformation mechanism

**Table 3 microorganisms-13-02880-t003:** Comparison of metabolomics analysis technology platforms.

Technical Platform	Detection Range and Characteristics	Typical Application Cases	Technical Advantages	Limitations
Gas chromatography–mass spectrometry (GC-MS)	A total of 665 effective peaks were detected, and 272 metabolites were identified, including amines, amino acids and organic acids. [95]	Standard Method for Analysis of Volatile Organic Compounds in Ruminant Research [95]	Electron bombardment ionization produces characteristic fragment patterns, which are suitable for the analysis of volatile organic compounds [95]	Limited detection of thermally unstable and non-volatile compounds [41]
Liquid chromatography–mass spectrometry (LC-MS)	The mass accuracy is less than 2 ppm, which is suitable for the accurate identification of polar metabolites [41]	Amino acid derivatives and bile acid compounds analysis [41]	High sensitivity and wide dynamic range, suitable for compounds with high polarity, large molecular weight, and poor thermal stability [41]	Matrix effect is obvious, which requires complex pretreatment [41]
Nuclear magnetic resonance (NMR)	A rumen fluid database containing 246 metabolites was constructed [96]	Dynamic metabolic process monitoring and non-destructive testing [96]	Non-destructive detection characteristics, suitable for dynamic process monitoring [96]	Low sensitivity and high detection limit [96]
Fourier transform infrared spectroscopy	Rapid acquisition of functional group information of metabolites [41]	And complementary analysis of mass spectrometry data [41]	Rapid detection and simple sample pretreatment [41]	have low specificity and are mainly used to assist identification [41]

**Table 4 microorganisms-13-02880-t004:** Influencing factors and mechanisms of rumen fermentation.

Categories of Influencing Factors	Specific Indicators	Degree of Influence/Range of Change	Key Role Mechanism
Diet factors [105]	Concentrate to forage ratio (30:70—70:30)	Acetic acid ↓ propionic acid ↑	Change microbial metabolic pathways.
Diet factors [105]	High-grain concentrate (>80%)	Methane conversion rate of 3–4%	Reduce methane to hydrogen source
Diet factors [105]	High fiber feed	Methane conversion rate of 10%+	Promote acetic acid/butyric acid fermentation
Diet factors [96]	Ionophore antibiotics	Microbial community function significantly changed	Regulating enzyme system activity
Environmental factors [106]	Temperature (25 °C–5 °C)	The amount of methane synthesis ↓	Fermentation mode to propionic acid
Environmental factors [106]	Humidity change	pH 6.8–5.8	Methanogenic archaea activity ↓ 78%
Environmental factors [101]	Redox potential (−350–−420 mV)	Cellulose decomposition rate ↑ 45%	Strictly anaerobic bacteria activity enhanced
Environmental factors [101]	Osmotic pressure (280–350 mOsm/kg)	Osmotic protectant production ↑ 2.6 times	Microbial homeostasis regulation
Host factor [7]	Methane emission in Holstein cows	Heritability 0.12–0.45	Genotype-specific regulation
Host factor [80]	Geographical distance (per 100 km)	The proportion of common flora ↓ 15%	Host genetic differentiation effect
Host factor [105]	Crossbreeding differences	18-month-old growth performance changes	Microbiome-metabolome recombination
Host factor	Physiological rhythm regulation	30% microbial group response	Multi-factor timing coordination

Note: ↑ indicates an increase, while ↓ indicates a decrease.

## Data Availability

No new data were created or analyzed in this study. Data sharing is not applicable to this article.

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
