# Peer review of "Mechanisms and Applications of Gastrointestinal Microbiota–Metabolite Interactions in Ruminants: A Review"

_microorganisms, 2025, doi:10.3390/microorganisms13122880_

Round 1

Reviewer 1 Report

Comments and Suggestions for Authors

The authors present an interesting retrospective analysis of the available literature on gastrointestinal microorganisms and metabolites in ruminants. Overall, the manuscript is well written. I have only one major comment: the authors should provide detailed information on the methodology used to systematically summarize the research status of microorganisms and metabolites in the ruminant gastrointestinal tract. Specifically, please describe the databases consulted, the total number of peer-reviewed studies included in the analysis, and the criteria used for study inclusion and exclusion. Additionally, the manuscript should clearly outline the systematic approach used to analyze and synthesize the collected information.

Author Response

Dear Reviewers and Editor:

On behalf of all the contributing authors, I would like to express our sincere appreciations of your letter and reviewers’ constructive comments concerning our article entitled “Mechanisms and Applications of Gastrointestinal Microbiota-Metabolite Interactions in Ruminants: A Systematic Review” (Manuscript No: microorganisms-4024656). These comments are all valuable and helpful for improving our article. According to the editor and reviewers’ comments, we have made modifications to our manuscript to make our results convincing. In the revised manuscript (with changes marked), we used the revision mode to mark the article for the convenience of editor and reviewers. Point-by-point responses to the nice editor and two nice reviewers are listed below this letter.

Reviewer #1:
The authors present an interesting retrospective analysis of the available literature on gastrointestinal microorganisms and metabolites in ruminants. Overall, the manuscript is well written. I have only one major comment: the authors should provide detailed information on the methodology used to systematically summarize the research status of microorganisms and metabolites in the ruminant gastrointestinal tract. Specifically, please describe the databases consulted, the total number of peer-reviewed studies included in the analysis, and the criteria used for study inclusion and exclusion. Additionally, the manuscript should clearly outline the systematic approach used to analyze and synthesize the collected information.

Thank you for your careful review and constructive feedback on this paper. These comments are all very valuable and helpful for improving our article. We have revised the article according to your request. Given the substantial changes made, we have used Track Changes for your convenience in reviewing the modifications. Thank you again for your valuable feedback, which has significantly contributed to enhancing the quality of the paper.

Sincerely,

Gerelt Zhao and co-authors

Corresponding author: Gerelt Zhao

Tel.: +086-13948317232; E-mail address: nmgrlt@imnu.edu.cn

College of Life Science and Technology, Inner Mongolia Normal University, Hohhot 010022, China.

Reviewer 2 Report

Comments and Suggestions for Authors

Comments on the Quality of English Language

The English writing is not adequate at this time.
The manuscript should be reviewed by a professional translation and writing service.

Author Response

The manuscript incorporates extensive and very recent literature (2021–2025), demonstrating the authors’ effort to include up-to-date findings. The tables are well designed and provide useful summaries that help structure a large amount of information.

However, there are several areas where the manuscript would benefit from substantial improvements. The current version includes repetitive content, lacks a clear conceptual framework, and would benefit from more critical synthesis rather than descriptive listings. In addition, parts of the text suffer from overly long sentences, redundancy, and minor issues in clarity and scientific style. The title also requires revision to better reflect the scientific content.

Title:

The use of the term “Inspiration” is unusual in scientific literature and does not accurately reflect the scope, depth, or analytical nature of this review. Given the comprehensive and mechanistic focus of the manuscript, a more descriptive title highlighting microbiota, metabolites, host interactions, and applied strategies would greatly improve clarity and alignment with the scientific content.

Thank you for your careful review and constructive feedback on this paper. These comments are all very valuable and helpful for improving our article. Based on your feedback, we have made extensive revisions to the manuscript within our capabilities. In this revised version, all modifications to our manuscript are highlighted in red text in the document. Based on your suggestions, the title has been revised to: "Mechanisms and Applications of Gastrointestinal Microbiota-Metabolite Interactions in Ruminants: A Systematic Review"

Abstract: The abstract is informative but could be improved by adding clearer structure about what is known, what remains unclear, what this review contributes and the practical or conceptual implications.

Thank you for your careful review and constructive feedback on this paper. These comments are all very valuable and helpful for improving our article. Based on your feedback, we have made extensive revisions to the manuscript within our capabilities. Based on your suggestions, the abstract has been revised to: "The gastrointestinal microbiota of ruminants constitutes a complex "invisible organ," which converts plant fibers into volatile fatty acids (VFAs) and microbial protein through fermentation, serving as the primary energy and protein sources for the host. While substantial progress has been made in characterizing this system, critical gaps remain in understanding causal mechanisms and translating knowledge into scalable interventions. This review systematically synthesizes current knowledge on the composition, function, and metabolite profiles of gastrointestinal microbial communities in ruminants, with emphasis on interaction mechanisms, methodological advances, and intervention strategies. We highlight persistent challenges, including the uncultured majority of microbes, causal inference limitations, and translational bottlenecks. The review further evaluates strategies for targeted microbiome modulation aimed at improving production performance and reducing environmental emissions. Finally, we propose integrated research priorities for developing efficient, low-carbon, and sustainable ruminant production systems."

Body of the manuscript:

The manuscript is rich in descriptive content but lacks critical evaluation in several major sections. - Introduction: Presents general background and known functions of the rumen microbiota, but does not identify research gaps, controversies, or methodological limitations.

Thank you for your careful review and constructive feedback on this paper. These comments are all very valuable and helpful for improving our article. Based on your feedback, we have made extensive revisions to the manuscript within our capabilities. Based on your suggestions, we have revised the Introduction section. The revised Introduction is as follows:

Global livestock production faces the dual challenges of enhancing production efficiency and reducing environmental impacts. In this context, studying the ruminant gut microbiota provides new insights into addressing this dilemma [1]. Ruminants rely on their unique digestive system and symbiotic intestinal microbial communities to efficiently convert plant fibers, which are inedible to humans, and agricultural by-products into high-quality animal proteins (such as meat and milk). This process is crucial for ensuring global food security and alleviating resource competition between humans and livestock [2,3]. However, translating microbial ecology knowledge into practical, reliable, and economically viable strategies remains a major bottleneck. This review aims not only to describe the current state of knowledge but also to critically analyze the mechanistic evidence, identify persistent knowledge gaps, and evaluate the practical potential and constraints of proposed intervention strategies.

Metagenomic studies have revealed that the ruminant gut microbiome represents one of nature’s most efficient biotransformation systems. Among the more than 5800 microbial genomes identified, over 90% of the species remain uncultured, and their functions are largely unknown [4], This "microbial dark matter" constitutes a major frontier and challenge for functional research. The research value of this field is mainly reflected in three dimensions: production – maximizing microbial efficiency in converting low-value feed into high-value protein [5,6]; environment – methane from ruminant microbial fermentation contributes approximately 5% of global anthropogenic greenhouse gas emissions, with a global warming potential 28 times that of carbon dioxide, rendering emission reduction critical for climate change mitigation [7–10]; and health – discoveries regarding rumen microbial involvement in bile acid metabolism and early-life gut colonization provide new models for studying human metabolic diseases and infant gut development [11,12]. Despite this potential, studies often report contradictory findings regarding microbial responses to dietary shifts, and the extrapolation of in vitro or small-scale trial results to heterogeneous field conditions is problematic.

The ruminant gastrointestinal microbial community functions not merely in digestion but as an essential "invisible organ" [13]. Its metabolites profoundly impact the host: short-chain fatty acids (SCFAs, such as acetic acid, propionic acid, and butyric acid) are not only core energy substrates, but also affect gene expression by regulating histone deacetylase activity; microbial derivatives of amino acids, such as tryptophan, can act as ligands to activate immune pathways and enhance intestinal barrier function [14]. Furthermore, microbially-mediated synthesis of B vitamins and vitamin K is essential for host metabolism and health [15]. A critical limitation is that most studies establish correlations rather than causality, and the precise concentration-dependent effects and tissue-specific actions of many metabolites in vivo are poorly quantified.

Innovations in research methodologies are central to progress in this field. From traditional culture methods to the combination of high-throughput technologies such as metagenomics, metabolomics, and stable isotope labeling, we can systematically analyze the structure and function of microbial communities and accurately track the transformation pathways of metabolites [16,17]. Nevertheless, each technology has inherent biases (e.g., PCR amplification bias, limitations in metabolite detection ranges, challenges in metagenome-assembled genome completeness) that must be considered when interpreting data. These technological advances have not only deepened our understanding of ruminant digestive physiology but also laid a foundation for developing microbiome-based precision feeding systems and transitioning animal production toward greater efficiency and environmental sustainability. Nevertheless, the field continues to face challenges in moving from descriptive phenomenology to mechanistic analysis and from single interventions to systemic regulation. This review systematically summarizes current research on microorganisms and metabolites in the ruminant gastrointestinal tract, focusing on the biosynthesis mechanisms and functional characteristics of microbial metabolites, elaborating their regulatory networks across host organs, critically evaluating existing intervention strategies, and outlining future research directions to advance novel microbiome-based strategies for ruminant health and production management.

- Section 2 (Digestive system and microbial ecology) – Provides detailed anatomical and functional descriptions without analyzing inconsistencies among studies, variability across breeds or environments, or limitations in current knowledge.

Thank you for your careful review and constructive feedback on this paper. These comments are all very valuable and helpful for improving our article. Based on your feedback, we have made extensive revisions to the manuscript within our capabilities. Based on your suggestions, we have revised the "Digestive system and microbial ecology" section. Specifically, we changed "While cellulose and hemicellulose are key fermentable substrates, it is important to note that ruminants possess superior fiber fermentation capacity compared to monogastric species like pigs and poultry." to "While cellulose and hemicellulose are key fermentable substrates, ruminants exhibit superior fiber fermentation capacity compared to monogastric species like pigs and poultry. It should be noted, however, that some monogastric species such as equines and rabbits can also ferment fiber appreciably in their ceca." Additionally, we changed "The rumen of newborn ruminants (such as calves and lambs and kids)…" to "The rumen of newborn ruminants (such as calves, lambs, and kids)…".

- Section 3 (Microbiome–metabolome interactions) – Lists studies and metabolic pathways sequentially, without comparing findings, discussing conflicting evidence, or evaluating the strength of the data.

Thank you for your careful review and constructive feedback on this paper. These comments are all very valuable and helpful for improving our article. Based on your feedback, we have made extensive revisions to the manuscript within our capabilities. Based on your suggestions, We have revised the "Microbiome–metabolome interactions" section. The updated content is as follows:

The microbial community in the intestinal tract of ruminants and the host metabolic system form a precise synergistic network. Bioactive molecules such as short-chain fatty acids, bile acid derivatives, and vitamins produced by microbial metabolism play key roles in energy supply and nutrient metabolism. These metabolites not only act as energy substrates but also affect the host's physiological state by regulating immune system function and metabolic pathway activity. It is noteworthy that changes in the host's metabolic environment also exert selective pressure on the composition of the microbial community, promoting adaptive changes in microbial population structure. This bidirectional regulation mechanism establishes a symbiotic relationship of dynamic balance between microorganisms and the host, which decisively influences the nutritional metabolic efficiency, energy conversion processes, and overall health status of ruminants. While this framework is widely accepted, the relative contribution of host genetics versus microbial activity to metabolic phenotypes, and the precise molecular triggers for microbial community shifts, are areas of active debate and require more causal evidence. This provides a new research paradigm for further analyzing the co-evolutionary relationship between hosts and microorganisms.

3.1 Microorganism-mediated metabolic regulation

The microbiome degrades cellulose, amino acids, and lipids to produce key metabolites (e.g., SCFAs, bile acids, vitamins), supplying up to 70% of the host’s energy [48,49]. Prevotella bryantii B14 activates the GPR109A–mTORC1 signaling pathway in the mammary gland via nicotinic acid metabolism, thereby regulating milk fat synthesis and revealing the mechanism underlying the "rumen–blood–mammary axis" [50]. In addition, virus-encoded auxiliary metabolic genes (AMGs) can expand host metabolic function and optimize nutrient utilization efficiency [51].

3.2 The feedback effect of metabolites on microorganisms

Microbial metabolites have a significant feedback regulation effect on microbial community structure. The accumulation of metabolites such as short-chain fatty acids and lactic acid can drive changes in the structure of the flora. For example, high-starch diets promote the enrichment of Succinivibrionaceae and change the flow of hydrogen metabolism, thereby affecting methane emissions [52,53]. The concentration of metabolites showed a dual effect. The appropriate concentration of butyrate promoted the development of rumen epithelium, while excessive butyrate led to epithelial parakeratosis [54].

3.3 Host-microorganism metabolic network

Host genetic background and microbiome together constitute a synergistic metabolic network. Bayesian model analysis showed that the host genetic factors explained about 24% of the variation in methane emissions, and the microbiome contributed about 7% [55]. Different intestinal types (such as TR and CO) have different microbial interaction patterns. TR intestinal type shows a more complex interconnection network and promotes efficient substrate utilization [50]. The interpretation of such network analyses requires caution, as correlation does not imply causation, and network stability and functional redundancy are rarely assessed.

3.4 Temporal and spatial distribution of microbial metabolites

Microbial metabolic activities showed significant spatial and temporal dynamic characteristics. In the rumen, 61.99% of the bacteria and 66.93% of the protozoa showed a circadian rhythm synchronized with the feeding time. The distribution of mobile genetic elements (MGEs) in the stomach and small intestine is characterized by "high at both ends and low in the middle". Among them, the stomach is rich in CAZyme carrying elements, and the ileum has the most resistance genes [56].

3.5 Host immune metabolic regulation

Microbial metabolites affect host health through immune regulation. Short-chain fatty acids promote the formation of anti-inflammatory immune cell phenotype and enhance intestinal barrier function by inhibiting HDAC activity and activating GPCR pathway [57]. Microbial tryptophan metabolite 3-indoleacetic acid (IAA) alleviates subacute rumen acidosis-related inflammation by inhibiting Th17 cell differentiation and IL-17 signaling pathway [58].

- Section 4 (Metabolite systems) – is highly descriptive, focusing on lists of metabolites and pathways. Missing are discussions of inter-individual variation, contradictory roles of certain metabolites, or challenges in metabolite quantification.

Thank you for your careful review and constructive feedback on this paper. These comments are all very valuable and helpful for improving our article. Based on your feedback, we have made extensive revisions to the manuscript within our capabilities. Based on your suggestions, We have revised the article. The updated content is as follows:

The intestinal microbial community of ruminants produces a variety of bioactive substances through complex metabolic pathways. These metabolites can be broadly categorized into primary and secondary metabolites based on their synthesis pathways and fundamental functions. They play key roles in host physiological processes. However, quantifying their absolute concentrations in different gut compartments and understanding inter-individual variation remains challenging. Table 2 systematically summarizes the types, producing flora, functional characteristics, and regulatory mechanisms of major intestinal metabolites in ruminants.

- Section 5 (Technological progress) – Enumerates methods and platforms, but does not address their limitations (e.g., PCR bias, MAG contamination, reproducibility issues, or constraints of predictive tools).

Thank you for your careful review and constructive feedback on this paper. These comments are all very valuable and helpful for improving our article. Based on your feedback, we have made extensive revisions to the manuscript within our capabilities. Based on your suggestions, We have revised the article, adding the following content to lines 352-356:

Nevertheless, these approaches possess inherent limitations. PCR amplification bias, incomplete metagenome-assembled genomes (MAGs), and variability in DNA extraction protocols can affect reproducibility. Furthermore, predictive metabolic models often lack experimental validation, and integrating multi-omics data remains computationally challenging.

- Section 6 (Influencing factors) – Describes dietary, environmental, and host factors without deeper analysis of inconsistencies, interactions among variables, or unresolved questions in the field.

- Section 7 (Control strategies) – Summarizes intervention approaches but does not critically discuss risks, field variability, regulatory limitations, cost feasibility, or comparative effectiveness.

Thank you for your careful review and constructive feedback on this paper. These comments are all very valuable and helpful for improving our article. Based on your feedback, we have made extensive revisions to the manuscript within our capabilities. Based on your suggestions, we have added a new subsection after “7.5 Epigenetic-based regulation,” with the following content:

7.6 Limitations and Practical Considerations of Intervention Strategies
While numerous strategies show promise in controlled settings, their translation to heterogeneous field conditions faces significant hurdles. Microbial adaptability, host genetic variability, and environmental interactions often attenuate intervention efficacy. Economic feasibility, regulatory approval, and consumer acceptance further constrain implementation. For instance, long-term use of feed additives may lead to microbial resistance or adverse effects on fiber digestion. Similarly, genetic editing and synthetic biology approaches raise ethical and biosafety concerns. Future research must prioritize robustness, scalability, and systemic impact assessment.

- Section 8 (Challenges and prospects) – Conceptually broad but general; lacks concrete prioritization of research needs, critical gaps, or specific limitations of current evidence.

Thank you for your careful review and constructive feedback on this paper. These comments are all very valuable and helpful for improving our article. Based on your feedback, we have made extensive revisions to the manuscript within our capabilities. Based on your suggestions, we have revised the “Challenges and Prospects” section as follows:

Although multi-omics technology has been widely used, it still faces challenges such as the analysis of unculturable microorganisms, the standardization of data integration, and the deeper application of artificial intelligence. A particularly important issue is the difficulty in directly comparing and reproducing many research results, due to inconsistencies in individual animal differences, feeding management, sample processing, and analytical procedures. Establishing standardized operating procedures (SOPs) and high-quality public databases is crucial.

The future innovation path should follow a dual-pronged approach:
(1) Breakthroughs in analytical technologies—focusing on in situ functional analysis. For example, single-cell multi-omics integration can link genetic potential with metabolic activity in uncultured microbes. Spatial multi-omics and organoid co-culture models can be used to visualize and verify the interaction mechanisms at the microbial–mucosal interface at the tissue level [143–145].
(2) Data- and model-driven advances—promoting the establishment of a global data alliance and a standardized database ecosystem [146]; applying AI-driven predictive modeling to understand microbiome responses to dietary or environmental changes and building a “digital twin of ruminants” system to provide the algorithmic core for precision nutrition [147,148]; and by incorporating synthetic biology principles, designing functional synthetic microbial communities while establishing a rigorous ecological safety assessment system to investigate their long-term impact on non-target communities, thereby advancing from “interpreting nature” to “safely designing function” [142,149].

This shift in research paradigm will advance ruminant microbiome studies from the traditional “production-oriented” approach to a new stage of multi-dimensional coordination among “ecology, production, health, and design.” It will not only provide core scientific and technological support for the sustainable development of animal husbandry but also offer new perspectives for understanding host–microorganism co-evolution, ultimately guiding the field into a new era of predictable, designable, and controllable precision regulation of the microbiome [150].

Several sections of the manuscript contain notable redundancy as well as overly long and complex sentences that affect readability and clarity. Content related to volatile fatty acids (VFAs), hydrogen metabolism, methane production, microbial fermentation, dietary effects, and host microbiome interactions is repeated across multiple sections. Redundancy is most evident in Sections 2.2, 3, 4, 6, 7, and 8, where similar explanations or concepts reappear without adding new perspectives or synthesis. It is recommend reducing redundancy by consolidating repeated concepts and ensuring that each section contributes new information or synthesis.

Thank you for your careful review and constructive feedback on this paper. These comments are all very valuable and helpful for improving our article. Based on your feedback, we have made extensive revisions to the manuscript within our capabilities. Based on your suggestions, we have revised the entire manuscript.

Numerous paragraphs include sentences exceeding 40–60 words, making them difficult to follow. This issue appears especially in the Introduction, Section 2.2, Sections 3 and 4, Section 5, and Section 8. These long constructions often combine several ideas (physiological, ecological, methodological, and applied) into a single sentence, reducing clarity and flow. You could break complex ideas into smaller units, and simplifying structure to significantly improve readability and overall scientific communication.

Thank you for your careful review and constructive feedback on this paper. These comments are all very valuable and helpful for improving our article. Based on your feedback, we have made extensive revisions to the manuscript within our capabilities. Based on your suggestions, we have revised the entire manuscript.

There is no conclusion in the manuscript. The conclusion must states clearly the key advances, identify major knowledge gaps, present concrete recommendations for future research and provide applied insights for sustainable ruminant production.

Thank you for your careful review and constructive feedback on this paper. These comments are all very valuable and helpful for improving our article. Based on your feedback, We have added a concluding section in the text, as follows:

This review synthesizes the intricate relationships between the gastrointestinal microbiota, their metabolites, and the ruminant host, highlighting both remarkable progress and persistent challenges. Key advances include genomic cataloging of rumen microbes, elucidation of major metabolic pathways (VFAs, hydrogen, nitrogen), and the recognition of microbial metabolites as central regulators of host energy metabolism, immunity, and health. The development of multi-omics and stable isotope tracing has transformed our observational capacity.

However, critical knowledge gaps remain. First, the field is rich in correlations but poor in causal mechanisms. The functional roles of the vast “uncultured majority” and the causal links between specific microbes, metabolites, and host phenotypes require rigorous validation using gnotobiotic models and synthetic communities. Second, research is imbalanced, heavily focused on few commercial species and the rumen, neglecting the hindgut’s role and the microbial adaptations of wild and indigenous ruminants. Third, translating promising interventions into robust, scalable, and economically viable on-farm strategies is a major bottleneck. Issues of microbial community stability, host adaptation, long-term efficacy, cost, and regulatory approval hinder implementation.

Future research must prioritize a holistic and mechanistic approach. We recommend: (1) expanding comparative studies across the ruminant phylogeny and integrating environmental microbiome data to understand fundamental principles of microbiome assembly and adaptation; (2) employing causal inference tools—germ-free models, temporal multi-omics, metabolic flux analysis—to move beyond association and define mechanism; (3) developing and validating "next-generation" interventions, including precision probiotics, engineered enzymes/microbes (with thorough safety assessments), and epigenetic modulators from microbial metabolites; and (4) embracing digital agriculture by integrating real-time sensor data (rumination, methane) with microbiome analytics for phenotyping and precision management.

Ultimately, harnessing the ruminant microbiome for sustainable production requires a paradigm shift from descriptive ecology to predictive biology and safe engineering. By addressing these priorities, targeted strategies can be developed to simultaneously enhance productivity, improve health and welfare, and mitigate environmental impact, contributing to a more resilient and sustainable global food system.

Other comments:

Some sections (e.g., Section 2 and Section 4) are dense and highly detailed, while others (e.g., those addressing limitations of technologies, ecological theory, or applied constraints) are relatively brief. Increasing balance by expanding underdeveloped areas (particularly discussions on limitations, controversies, and comparative perspectives) would improve coherence across the manuscript.

Thank you for your careful review and constructive feedback on this paper. These comments are all very valuable and helpful for improving our article. Based on your feedback, we have made extensive revisions to the manuscript within our capabilities. Based on your suggestions, we have revised the entire manuscript.

Although the manuscript is extensive and content-rich, it currently lacks conceptual figures. Including visual summaries (e.g., microbiome–metabolome–host interaction maps, hydrogen metabolic networks, or a schematic of intervention strategies) would greatly enhance clarity and reader engagement.

Thank you for your careful review and constructive feedback on this paper. These comments are all very valuable and helpful for improving our article. We sincerely apologize. We did attempt to create conceptual figures, but the results were not satisfactory, so we decided not to include them. We regret this omission.

In Tables 1 and 2, scientific names (genus and species) must be in italics.

Table 4 lacks capital letters at the beginning of most statements in columns 2,3, and 4

There is an overuse of quotation marks ("") and the English writing is not suitable for a review as it does not show continuity of ideas.

Thank you for your careful review and constructive feedback on this paper. These comments are all very valuable and helpful for improving our article. Based on your feedback, We have corrected the italicization and capitalization in the tables, reviewed the usage of quotation marks, and made the necessary revisions.

Reviewer 3 Report

Comments and Suggestions for Authors

General comments:

This narrative review addresses a holistic approach to microbiome modulation regarding the improvement of ruminants' health and the mitigation of greenhouse gas emissions. This is a relevant subject within the scope of the Microorganisms journal. The ruminal microbiota and metabolites were deeply described and metabolomics analysis technologies to support research were compared. The main factors to modulate the ruminal microbiome are assessed, as well as strategies to control them. A deeper review (current state-of-the-art) of “Innovative research methods” is welcome. The aim is to give some specific ideas with potential expected outcomes. A separate conclusion is also welcome. Despite these issues, this review is well structured and written with a high level of readability. The introduction provides a detailed background of this topic. The successive subsections provide a complete overview and are supported by a large amount (150) of relevant literature. All four tables are elucidative and informative.  

Specific comments:

L102 (Table 1): Please italicize when appropriate.

L111-115:   Cellulose and hemicellulose are fibers that present low ruminal fermentation when compared with others.  The ruminants are better species than monogastrics like pigs and poultry for fiber fermentation in the rumen (equine and rabbits can ferment appreciable fiber in the cecum, and are monogastrics).

L150: and kids (goats).

L221 (Table 2): Please italicize when appropriate.

L529: Conclusions are welcome.

Author Response

This narrative review addresses a holistic approach to microbiome modulation regarding the improvement of ruminants' health and the mitigation of greenhouse gas emissions. This is a relevant subject within the scope of the Microorganisms journal. The ruminal microbiota and metabolites were deeply described and metabolomics analysis technologies to support research were compared. The main factors to modulate the ruminal microbiome are assessed, as well as strategies to control them. A deeper review (current state-of-the-art) of “Innovative research methods” is welcome. The aim is to give some specific ideas with potential expected outcomes. A separate conclusion is also welcome. Despite these issues, this review is well structured and written with a high level of readability. The introduction provides a detailed background of this topic. The successive subsections provide a complete overview and are supported by a large amount (150) of relevant literature. All four tables are elucidative and informative.

Specific comments:

L102 (Table 1): Please italicize when appropriate.

L221 (Table 2): Please italicize when appropriate.

Thank you for your careful review and constructive feedback on this paper. These comments are all very valuable and helpful for improving our article. Based on your feedback, We have corrected the italic formatting in Table 1 and Table 2. The revised version is as follows:

Table 1. Distribution and functional characteristics of microbial communities in the digestive tract of ruminants.

Digestive tract parts.

Dominant flora/functional flora

Cell density/abundance

Main functions

Influencing factors

Rumen

PrevotellaRuminococcusButyrivibrio [28]

10^10-10^11cells/mL

[28]

Fiber degradation, nitrogen metabolism, and volatile fatty acid synthesis [28]

Diet composition, host genetics [28,29]

Rumen

Fibrobacter succinogenesRuminococcus albus [30]

-

Degradation of cellulose and hemicellulose [30]

Dietary fiber content [30]

Rumen

Methanobrevibacter [28]

It accounts for 0.3-3% of the microbiome [28]

methane production [28]

hydrogen availability [28]

Small intestine

LactobacillusStreptococcus, Enterobacteriaceae etc [31]

-

Oligopeptides and amino acid absorption, subsequent digestion of residual starch and protein, immune regulation [31]

Amino acid composition [31]

Large intestine

BacteroidesClostridiumFibrobacter etc

[32, 33]

-

Undigested fiber and protein fermentation, SCFAs secondary synthesis, water absorption [33]

Dietary residues [33]

Rumen liquid phase

Succinivibrio dextrinosolvens [34]

-

Pectin and maltose are degraded to produce succinic acid and acetic acid [34]

soluble carbohydrates [34]

Rumen solid phase

Fibrobacter succinogenes [30]

-

Crystalline cellulose degradation [30]

fiber substrate [30]

ruminal epithelium

-

-

Epithelial cell regulation [28]

host-microorganism interaction [28]

Digestive tract of newborn individual

Mother-derived microorganisms [35]

-

Early colonization [35]

Maternal exposure and environmental exposure [35]

Table 2. Metabolites and functional characteristics of gut microbiota in ruminants.

Metabolite type

mainly producing flora

main functional characteristics

concentration/ratio characteristics

key regulatory mechanisms

Short-chain fatty acids (SCFAs)

Firmicutes (butyric acid), Bacteroidetes (acetic acid/propionic acid) [59]

Energy supply (meeting 50% of rumen energy demand) [60], immune regulation (GPR41/43 pathway) [61], intestinal barrier function

Proximal large intestine 57:22:21(acetic acid: propionic acid: butyric acid) [62]

Activation of G protein-coupled receptors, inhibition of HDAC [63], inhibition of NF-κB signaling [64], and promotion of Treg cell proliferation [65]

Bile acid

Clostridium (such as Clostridium scindens) [48, 68]

lipid digestion and absorption, GLP-1 secretion promotion [62], metabolic regulation (FXR/GPBAR1 receptor) [12]

High starch diet increases TCDCA/TDCA levels[66]

7-dehydroxylation transformation[59], FXR nuclear receptor activation [18], TGR5 receptor-mediated energy expenditure [62]

Tryptophan derivatives

Bifidobacterium et al. [59]

Immunomodulatory (IL-22 induction) [12],neurotransmitter synthesis (5-HT) [67],inflammatory regulation (Th17 inhibition) [68]

Increased metabolic activity in IBD patients

AhR receptor activation [23], IDO enzyme expression upregulation [59], KYN pathway metabolite regulation [12]

Polyamines

PrevotellaRuminococcus

Epithelial cell proliferation promotion, autophagy regulation, and anti-inflammatory effects (IL-10 promotion/TNF-α inhibition)

It accounts for 30-40% of intestinal polyamine pool

Arginine decarboxylase/ornithine decarboxylase pathway, HDAC inhibition, macrophage polarization regulation

Vitamins

Bacteroidetes, Firmicutes

Nutrition supply (B vitamins), coagulation function (VitK), immune regulation (Foxp3+ T cell homeostasis)

[59]

Nicotinic acid inhibits IL-8 production [12]

Epigenetic modification is involved in the regulation of inflammatory signaling pathways

Choline metabolites

Specific intestinal flora

Cardiovascular health indicators, lipid metabolism regulation

The content of TMA decreased in IBD patients [12]

Choline-TMA-TMAO metabolic axis and liver transformation mechanism

L111-115: Cellulose and hemicellulose are fibers that present low ruminal fermentation when compared with others. The ruminants are better species than monogastrics like pigs and poultry for fiber fermentation in the rumen (equine and rabbits can ferment appreciable fiber in the cecum, and are monogastrics).

Thank you for your careful review and constructive feedback on this paper. These comments are all very valuable and helpful for improving our article. Based on your feedback, As per your suggestions, we have revised the text from: "Rumen microorganisms produce volatile fatty acids (VFAs), mainly including acetic acid, propionic acid and butyric acid, by fermenting carbohydrates (mainly cellulose and hemicellulose) in the diet. After being absorbed by the rumen wall, these VFAs can meet 70%-80% of the maintenance and growth energy needs of ruminants." to: "Rumen microorganisms produce volatile fatty acids (VFAs), mainly including acetic acid, propionic acid and butyric acid, by fermenting carbohydrates (mainly cellulose and hemicellulose) in the diet. While cellulose and hemicellulose are key fermentable substrates, ruminants exhibit superior fiber fermentation capacity compared to monogastric species like pigs and poultry. It should be noted, however, that some monogastric species such as equines and rabbits can also ferment fiber appreciably in their ceca."

L150: and kids (goats).

Thank you for your careful review and constructive feedback on this paper. These comments are all very valuable and helpful for improving our article. Based on your feedback, We have revised the paper.

L529: Conclusions are welcome.

Thank you for your careful review and constructive feedback on this paper. These comments are all very valuable and helpful for improving our article. Based on your feedback, We have added a concluding section in the text, as follows:

This review synthesizes the intricate relationships between the gastrointestinal microbiota, their metabolites, and the ruminant host, highlighting both remarkable progress and persistent challenges. Key advances include genomic cataloging of rumen microbes, elucidation of major metabolic pathways (VFAs, hydrogen, nitrogen), and the recognition of microbial metabolites as central regulators of host energy metabolism, immunity, and health. The development of multi-omics and stable isotope tracing has transformed our observational capacity.

However, critical knowledge gaps remain. First, the field is rich in correlations but poor in causal mechanisms. The functional roles of the vast “uncultured majority” and the causal links between specific microbes, metabolites, and host phenotypes require rigorous validation using gnotobiotic models and synthetic communities. Second, research is imbalanced, heavily focused on few commercial species and the rumen, neglecting the hindgut’s role and the microbial adaptations of wild and indigenous ruminants. Third, translating promising interventions into robust, scalable, and economically viable on-farm strategies is a major bottleneck. Issues of microbial community stability, host adaptation, long-term efficacy, cost, and regulatory approval hinder implementation.

Future research must prioritize a holistic and mechanistic approach. We recommend: (1) expanding comparative studies across the ruminant phylogeny and integrating environmental microbiome data to understand fundamental principles of microbiome assembly and adaptation; (2) employing causal inference tools—germ-free models, temporal multi-omics, metabolic flux analysis—to move beyond association and define mechanism; (3) developing and validating "next-generation" interventions, including precision probiotics, engineered enzymes/microbes (with thorough safety assessments), and epigenetic modulators from microbial metabolites; and (4) embracing digital agriculture by integrating real-time sensor data (rumination, methane) with microbiome analytics for phenotyping and precision management.

Ultimately, harnessing the ruminant microbiome for sustainable production requires a paradigm shift from descriptive ecology to predictive biology and safe engineering. By addressing these priorities, targeted strategies can be developed to simultaneously enhance productivity, improve health and welfare, and mitigate environmental impact, contributing to a more resilient and sustainable global food system.

Round 2

Reviewer 2 Report

Comments and Suggestions for Authors

The authors have made a clear effort to address the reviewers’ comments, and most aspects of the manuscript have improved, particularly the title, abstract structure, and the inclusion of methodological limitations. The revised version shows progress toward a more analytical perspective compared to the original submission. 

In tables 1, 2 and 3 there is still information that should begin with a capital letter.

Comments on the Quality of English Language

No comments.